# Biochemical and Neuropathological Findings in a Creutzfeldt–Jakob Disease Patient with the Rare Val180Ile-129Val Haplotype in the Prion Protein Gene

**DOI:** 10.3390/ijms231810210

**Published:** 2022-09-06

**Authors:** Gianluigi Zanusso, Elisa Colaizzo, Anna Poleggi, Carlo Masullo, Raffaello Romeo, Sergio Ferrari, Matilde Bongianni, Michele Fiorini, Dorina Tiple, Luana Vaianella, Marco Sbriccoli, Flavia Porreca, Michele Equestre, Maurizio Pocchiari, Franco Cardone, Anna Ladogana

**Affiliations:** 1Department of Neurosciences, Biomedicine and Movement Sciences, University of Verona, 37134 Verona, Italy; 2Department of Neuroscience, Istituto Superiore di Sanità, 00161 Rome, Italy; 3Department of Neuroscience, Catholic University of the Sacred Heart, 00168 Rome, Italy; 4Unit of Otolaryngology, Clinica Nuova Itor Rome, 00158 Rome, Italy

**Keywords:** Creutzfeldt–Jakob disease, prion disease, neuropathology, prion protein, Val180Ile mutation, RT-QuIC assay

## Abstract

Genetic Creutzfeldt–Jakob disease (gCJD) associated with the V180I mutation in the prion protein (PrP) gene (*PRNP*) in phase with residue 129M is the most frequent cause of gCJD in East Asia, whereas it is quite uncommon in Caucasians. We report on a gCJD patient with the rare V180I-129V haplotype, showing an unusually long duration of the disease and a characteristic pathological PrP (PrP^Sc^) glycotype. Family members carrying the mutation were fully asymptomatic, as commonly observed with this mutation. Neuropathological examination showed a lesion pattern corresponding to that commonly reported in Japanese V180I cases with vacuolization and gliosis of the cerebral cortexes, olfactory areas, hippocampus and amygdala. PrP was deposited with a punctate, synaptic-like pattern in the cerebral cortex, amygdala and olfactory tract. Western blot analyses of proteinase-K-resistant PrP showed the characteristic two-banding pattern of V180I gCJD, composed of mono- and un-glycosylated isoforms. In line with reports on other V180I cases in the literature, Real-Time Quaking Induced Conversion (RT-QuIC) analyses did not demonstrate the presence of seeding activity in the cerebrospinal fluid and olfactory mucosa, suggesting that this haplotype also may result in a reduced seeding efficiency of the pathological PrP. Further studies are required to understand the origin, penetrance, disease phenotype and transmissibility of 180I-129V haplotype in Caucasians.

## 1. Introduction

Genetic Creutzfeldt–Jakob disease (gCJD) is a Mendelian inherited neurodegenerative disorder affecting humans and belonging to prion diseases [1]. In addition to gCJD, human prion diseases (PD) include other genetically transmitted disorders (Gerstmann–Sträussler–Scheinker disease and fatal familial insomnia), idiopathic disorders (sporadic CJD, sporadic fatal insomnia, and variably protease-sensitive prionopathy) and acquired disorders (iatrogenic CJD and variant CJD). The major pathogenic event of PD is the misfolding of the glycosylated cellular prion protein (PrP^C^) into pathological conformers containing high levels of beta-chains (PrP^Sc^) [2,3]. Both the cellular and misfolded prion proteins contain unglycosylated, mono- and di-glycosylated isoforms that run in three distinct bands at the electrophoresis. However, the proteinase K (PK) treatments completely digest PrP^C^ while they only trim the N-terminus of PrP^Sc^, yielding, in human PD, two major types of partially digested PrP^Sc^ (PrP27–30) depending on the molecular weight of the unglycosylated isoform: approximately 21 kDa for type 1 and 19 kDa for type 2 [4].

The mutations in the prion protein gene (*PRNP*) coding sequence are linked to the genetic forms of PD, likely facilitating PrP^Sc^ production, deposition and accumulation [5]. *PRNP* mutations consist of point mutations leading to amino acid substitution, premature stop codon or the insertion of octapeptide repeats in the N-terminal region of the *PRNP* [6]. PrP27–30 types, mutations and the presence of methionine or valine at the polymorphic position 129 of the *PRNP* are the main determinants for clinical, pathological and biochemical phenotypes of human PD.

Here, we report on a patient with prion disease of an unusually long duration with clinical features of sporadic CJD (sCJD) who carried the V180I-129V haplotype and showed a unique pattern of PrP^Sc^ in the Western blot.

## 2. Results

### 2.1. Case Report

The patient, a 74-year-old Caucasian man of Italian origin, developed psychiatric symptoms (mood depression, insomnia, apathy, abulia and withdrawal) after a period of emotional stress treated with antidepressants without benefit. One month later, the patient’s condition worsened with the appearance of forgetfulness, ideomotor apraxia, dysphasia, agraphia, unsteady gait and falls. Seven months later, he was hospitalized because of the rapid progression in cognitive and motor symptoms. At this time, the Mini-Mental State Examination (MMSE) score was 9/30. During hospitalization, the patient developed myoclonus in the upper limbs, extrapyramidal and pyramidal signs and global aphasia. CJD was suspected. Several EEG recordings were performed showing diffuse slowing with generalized theta–delta activity without typical periodisms. Fluid-attenuated inversion recovery (FLAIR) and diffusion-weighted (DWI) magnetic resonance imaging (MRI) sequences of the brain showed moderate and bilateral hyperintensity of the basal ganglia (caudate and putamen) and a striking bilateral cortical hyperintensity, more evident in the frontal and parietal-temporal regions (Figure 1).

The cerebrospinal fluid (CSF) analyses showed low levels of total tau proteins and no 14-3-3. After obtaining informed consent from the relatives, a blood sample was collected, and genomic DNA was extracted from peripheral leukocytes according to standard procedures. *PRNP* analysis revealed that the patient was heterozygous for G to A (GTC to ATC) at codon 180, resulting in the missense amino acid substitution of valine to isoleucine. The genotype at the 129 polymorphic site was homozygous for valine. The patient was discharged from the hospital and followed at home, where he remained stable for several months. Eighteen months after the disease onset, the patient underwent nasal brushing to collect the olfactory mucosa. Six months later (24 months after onset), he had liquid dysphagia and was bedridden. He died 35 months after disease onset. An autopsy (limited to the brain) was performed.

The patient’s past medical history was overall unremarkable, apart from surgical interventions, consisting of a nephrectomy and a cholecystectomy 12 and 3 years before onset, respectively. The family history was negative for dementia or other neurological disorders.

Two (65 and 45 years) of the three healthy family members who asked for *PRNP* testing during genetic counseling carried the V180I mutation.

### 2.2. Real-Time Quaking Induced Conversion

Real-Time Quaking Induced Conversion (RT-QuIC) analyses performed on CSF and olfactory epithelium (OE) samples taken, respectively, 7 and 18 months after onset (Figure 2) did not reveal the presence of newly formed Thioflavin T (ThT) reactive aggregates, even though different recombinant prion proteins were used as substrates for the reaction.

### 2.3. Neuropathological Changes

All cortical layers showed non-confluent vacuoles of a different size, mainly assembled in the deeper layers. Mild spongiosis with highly reactive gliosis was prominent in the frontal cortex throughout all cortical layers (Figure 3A). More severe spongiosis was observed in the occipital cortex (Figure 3B). PrP immunostaining of the cerebral cortexes was characterized by punctate synaptic-like PrP deposition with rare PrP aggregates (Figure 3C,D). In the entorhinal cortex, glial-like PrP deposits were occasionally observed (Figure 3D,D’,G). In contrast, no pathological changes were observed in the subcortical white matter. Moderate spongiosis was observed in the hippocampus, particularly in the dentatus gyrus. The amygdala was severely affected, and PrP immunostaining showed a synaptic-like pattern of deposition with sporadic intracellular deposits (Figure 3E,E’). The thalamus and striatum were mildly affected, albeit in the thalamus intracellular PrP deposits (Figure 3F,H), and sporadic PrP aggregates (Figure 3I) were observed. Olfactory brain areas, including the pre-pyriform frontal and temporal areas, were severely affected. Moreover, PrP immunostaining was present in the olfactory bulb and in the olfactory tract. In contrast, the cerebellum and nucleus dentatus were completely spared.

No aging pathology was observed, including senile plaques or neurofibrillary tangles. The amyloid-beta immunostaining was negative, while the tau immunostaining showed sporadic positive neurons and neurites in the hippocampus (Figure 3J).

### 2.4. Immunoblot Analysis and Regional Distribution of Protease-Resistant PrP

An immunoblot analysis with a 3F4 antibody showed the absence of the di-glycosylated PrP^Sc^ isoform and the presence of a distinctive minor 8 kDa band (Figure 4A) in the brain samples from the V180I CJD patient. Proteinase-K digested brain samples (Figure 4B) showed the presence of PrP^Sc^, characterized by two main bands migrating at 20 (unglycosylated isoform) and 25 kDa (monoglycosylated isoform). The pattern of the PrP^Sc^ glycotype was identical in all the examined brain areas. The highest levels of PrP^Sc^ were observed in the entorhinal and anterior cingulate cortexes, while the lowest levels were in the hippocampus (Figure 4B).

Immunostaining of PrP^Sc^ with antibody SA21 displays the same two bands detected by the 3F4 mAb and confirms only the presence of unglycosylated and 197-monoglycosylated PrP^Sc^ isoforms.

## 3. Discussion

The genetic forms of human PD occur with variable frequencies in different populations [7]. A striking example is genetic V180I CJD that is extremely rare in the Caucasian population [7,8,9], while it is the most frequent *PRNP* mutation in East Asia [10,11,12]. The case described here is the only case of gCJD associated with a V180I mutation reported in Italy since the establishment of the national CJD surveillance in 1993. The case of the patient reported in Brazil who had Italian ancestry [9] does not belong to the same family as our case because she carried 129M instead of 129V in the mutant allele. The unprecedented finding of the V180I mutation in coupling with the valine residue in position 129 supports the absence of any relationship with other published Caucasian cases, advocating for an independent G to A transition as a possible origin for this mutation.

Although the V180I frequency in CJD patients remarkably varies between Caucasian and Oriental populations, it rarely occurs in more than a single member of the affected families. Thus, the V180I mutation is considered to have very low penetrance [13,14] and modest pathogenic impact in line with molecular simulation studies showing a negligible effect of V180I on the stability of PrP [15]. Similarly to the Japanese cases, the Italian case had no family history of CJD or other rapidly progressive dementias, although the relatively young age of the two healthy family members carrying the mutation does not allow for the formulation of any strong hypothesis on the penetrance of the V180I-129V mutant allele.

The clinical presentation of this patient was remarkably similar to that described in Japanese patients [13,16] with psychiatric and cognitive features at onset and a long disease duration. Only the brain MRI supported the intra vitam diagnosis of CJD according to international criteria [17,18]. The brain pathology also revealed concordance with the Japanese cases in the severity of lesions (spongiform change) in the cerebral cortex [19,20]. The only difference was the presence of moderate spongiosis in the hippocampus and the severe impairment of the amygdala in the Italian patient compared to the Japanese cases [19,20]. The cerebellar cortex was spared, as previously reported and in line with the absence of cerebellar signs. The white matter was also spared in spite of prolonged disease duration.

Overall, the neuropathological lesions were similar to those reported in the Japanese cases carrying the mutation coupled with methionine [13,19]. It is then possible to speculate that the clinico-pathological features in V180I patients are mainly governed by the mutant allele with apparently no influence from the M/V residue in position 129 of the mutant allele.

Likewise, the characteristic immunoblot PrP27–30 pattern in which the di-glycosylated band is missing and the unglycosylated band migration is slightly slower than type 2 PrP^Sc^ [20,21] is likely related to the substitution of valine with isoleucine at codon 180, which might influence the PrP^Sc^ PK-cleavage at the N-terminus. The absence of the di-glycosylated PrP^Sc^ after PK treatment has been also observed in variably protease-sensitive prionopathy and in rare cases of atypical sCJD [21,22,23,24] and is attributable to a glycoform-selective PrP^C^ to PrP^Sc^ conversion process rather than to a primary absence of the precursor PrP^C^ isoforms [21], as observed in patients with gCJD associated with the T183A mutation.

Interestingly, PrP^Sc^ was detectable in samples from the olfactory bulb and the olfactory tract, but RT-QuIC seeding activities in the OE and CSF were always negative with all recombinant PrPs used as the substrate for replication. A negative RT-QuIC in CSF was observed in approximately 40% of Japanese patients with V180I-129M [13,25] and in one of four Korean patients [11], making it difficult to relate our negative results to the presence of valine 129 in the mutant allele. It is likely that the V180I mutant PrP^Sc^ has an intrinsic poor efficiency for seeding [24]. On the other hand, RT-QuIC seeding activity in the OE was never reported in V180I patients, leaving uncertainties in the meaning of this result. Possibly, the amount of mutant V180I PrP^Sc^ in olfactory neurons is very low or absent.

Further studies are needed to better understand the overall mechanisms of the disease in carriers of the rare 180I-129V haplotype. The origin of the mutation may be investigated by NGS sequencing of exon 2 in the *PRNP* gene to identify specific haplotypes [26]; an estimate of V180I penetrance in rare diseases is possible only through collaboration with the European surveillance network in order to include as many cases as possible with accurate clinical and laboratory data. Finally, it is important to determine whether this rare form of gCJD is transmissible to laboratory animals. The data from Japanese researchers are not yet conclusive for V180I-129M [27] and no research has been performed with V180I-129V material. We are planning to attempt transmission by inoculating the highly susceptible bank voles with the brain homogenate from our case [28] to determine any eventual risk of iatrogenic transmission.

## 4. Materials and Methods

### 4.1. Real-Time Quaking Induced Conversion

An RT-QuIC assay was performed to detect the presence of the pathological misfolded PrP in CSF and the OE collected by nasal brushing, using standard protocols in the presence of the recombinant full-length hamster prion protein (Ha rPrP 23–231) as a substrate for the reaction [29,30]. Moreover, additional tests were run with either the recombinant, truncated hamster prion protein (Ha rPrP 90–231) or the recombinant bank vole full-length prion protein (BV rPrP 23–230). Recombinant PrP substrates were prepared in-house. All tests were carried out in quadruplicate by seeding 30 µL of undiluted CSF and 2 µL of OE (10^−3^ and 10^−4^ dilutions) on a FluoStar OMEGA plate reader (BMG LABTECH, Ortenberg, Germany), at 42 °C or 50 °C, as previously reported [31,32]. Briefly, RT-QuIC analyses with the Ha rPrP 23–231 and BV rPrP 23–230 substrates were performed using a standard 10 mM phosphate buffer (pH  7.4), 170 mM NaCl (total 400 mM including phosphate buffer) containing 0.1 mg/mL rPrP, 10 μM ThT, and 10 µM ethylenediaminetetraacetic acid tetrasodium salt and incubated at 42 °C. For the Ha rPrP 90–231 substrate, 0.002% SDS (sodium dodecyl sulfate) was incorporated, and incubation was carried out at 50 °C. The final volume for all substrates was 100 μL.

Plates were sealed and incubated with intermittent shaking cycles consisting of 90 sec double orbital shaking at 900 rpm followed by 30 s rest for the Ha rPrP 23–231 recombinant protein, while for the other two substrates, the intermittent shaking cycles consisted of 1 min shaking at 700 rpm followed by 1 min rest. The ThT fluorescent signal (450 nm excitation and 480 nm emission) was measured every 15 min in relative fluorescence units (rfu). We considered samples to be positive when at least two replicates out of four exceeded the minimum reading of that well plus 10% of the maximum reading on the plate.

### 4.2. Neuropathological Study

During the autopsy, the left half of the brain was fixed in formalin for neuropathological examination. The specimens obtained for pathological examination were pretreated with formic acid for one hour before embedding in paraffin. The contralateral areas were frozen and stored at −80 °C for biochemical analyses. The formalin-fixed sections of the cerebral hemispheres, cerebellum and brainstem were subjected to histological analyses, including hematoxylin-eosin (H&E) staining.

For PrP^Sc^ immunohistochemistry, the brain sections were processed after hydrolytic autoclaving. The sections were deparaffinized, rehydrated and immersed in 98% formic acid for 1 h at room temperature. Endogenous peroxidase was blocked by immersion in 8% hydrogen peroxide in methanol for 10 min. To unmask PrP^Sc^ epitopes, the sections were processed in 1.5 mM HCl and autoclaved at 121 °C for 10 min. After rinsing, slides were incubated overnight with the anti-prion 3F4 monoclonal antibody (1:1000, Covance, Princeton, NJ, USA, catalog# SIG-39620) diluted and with anti-tau antibody AT8 (1:1000, Thermofisher, Waltham, MA, USA, catalog# MN1020). Sections were subsequently incubated with a biotinylated goat anti-mouse antibody (1:200, Fisher Scientific Italia, Rodano, Italy, catalog# RPN1001-2ML) for 1 h at room temperature and then with the avidin–biotin solution (ABC-Elite Kit, Vector lab, Burlingame, CA, USA) according to the manufacturer’s instructions, developed with 3,3′-diaminobenzidine (Merck, Kenilworth, NJ, USA) and counterstained with hematoxylin.

### 4.3. Western Blot Analysis

Western blot analysis was carried out on frozen samples of the different areas of the cerebral cortex. Briefly, brain tissue samples were homogenized in 9 volumes of lysis buffer (100 mM sodium chloride, 10 mM EDTA, 0.5% Nonidet P-40, 0.5% sodium deoxycholate, 10 mM Tris, pH 7.4) and clarified by centrifugation at 1000× *g* for 10 min. Protease resistance was assayed by a final concentration of 50 μg/mL of proteinase K (Roche Applied Science, Penzberg, Germany) and incubated at 37 °C for 60 min. The reaction was stopped by the addition of a protease inhibitor (5 mM phenylmethylsulfonyl fluoride, Merck Life Science, Milano, Italy). Proteins were then dissolved in the sample buffer (3% SDS, 3% β-mercaptoethanol, 2 mM EDTA, 10% glycerol, 62.5 mM Tris, pH 6.8, Merck) and boiled for 5 min. An equivalent of 0.4 mg of wet tissue was loaded on 13% SDS-PAGE gels, and the proteins were transferred onto PVDF membrane (Immobilon P, Millipore, Merck Life Science, Milano, Italy) for 2 h at 60 V. Membranes were blocked with 1% non-fat dry milk in TBST (10 mM Tris, 150 mM NaCl, 0.1% Tween-20, pH 7.5) for 1 h at 37 °C and incubated overnight at 4 °C with anti-PrP monoclonal antibodies 3F4 and SA21 [33]. According to the producer, 3F4 (1:5000, Covance, Princeton, NJ, USA, catalog# SIG-39620) recognizes the human PrP sequence between 109 and 112, while our SA21 (1:1000) distinguishes between residues 165 and 185 of both unglycosylated PrP and PrP glycosylated at position 197, but it does not recognize PrP isoforms glycosylated at 181 [33].

After 1 h incubation with secondary anti-mouse IgG HRP Linked Whole Antibody (1:3000 dilution, Fisher Scientific, Italia, Rodano, Italy, catalog# NA931-1ML), blots were developed by an enhanced chemiluminescence system (ECL, GE Healthcare, Chicago, IL, USA) and PrP visualized by chemiluminescence Imager (ALLIANCE Q9-MICRO, Uvitec, Cambridge, UK).

## 5. Conclusions

We report detailed clinical, laboratory, genetic and neuropathology data from a Caucasian patient presenting with a CJD phenotype and carrying the unique combination of the V180I mutation in phase with valine at the polymorphic codon 129. Our case has striking similarities with patients carrying the more common V180I-129M mutation, suggesting that the polymorphic residue at position 129 of the mutated allele does not affect the disease phenotype. These findings are relevant for future works that aim to understand the relationship between the protein sequence, misfolding patterns and disease phenotype.

## Figures and Tables

**Figure 1 ijms-23-10210-f001:**
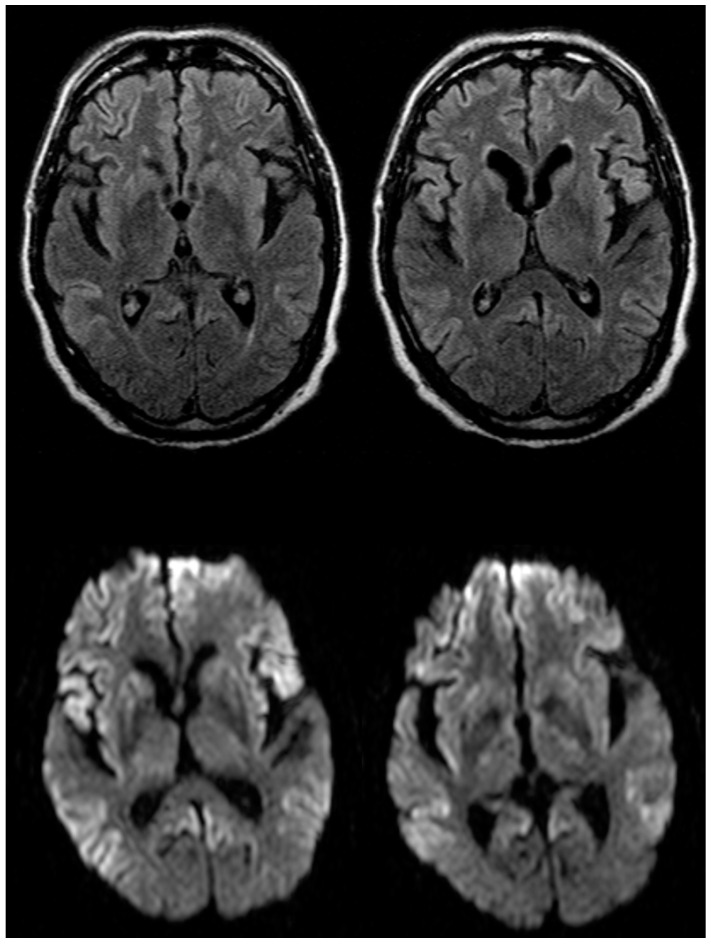
MRI scans from patient V180I. Upper row: fluid attenuated inversion recovery images at the level of the basal ganglia; lower row: diffusion weighted images at the same levels.

**Figure 2 ijms-23-10210-f002:**
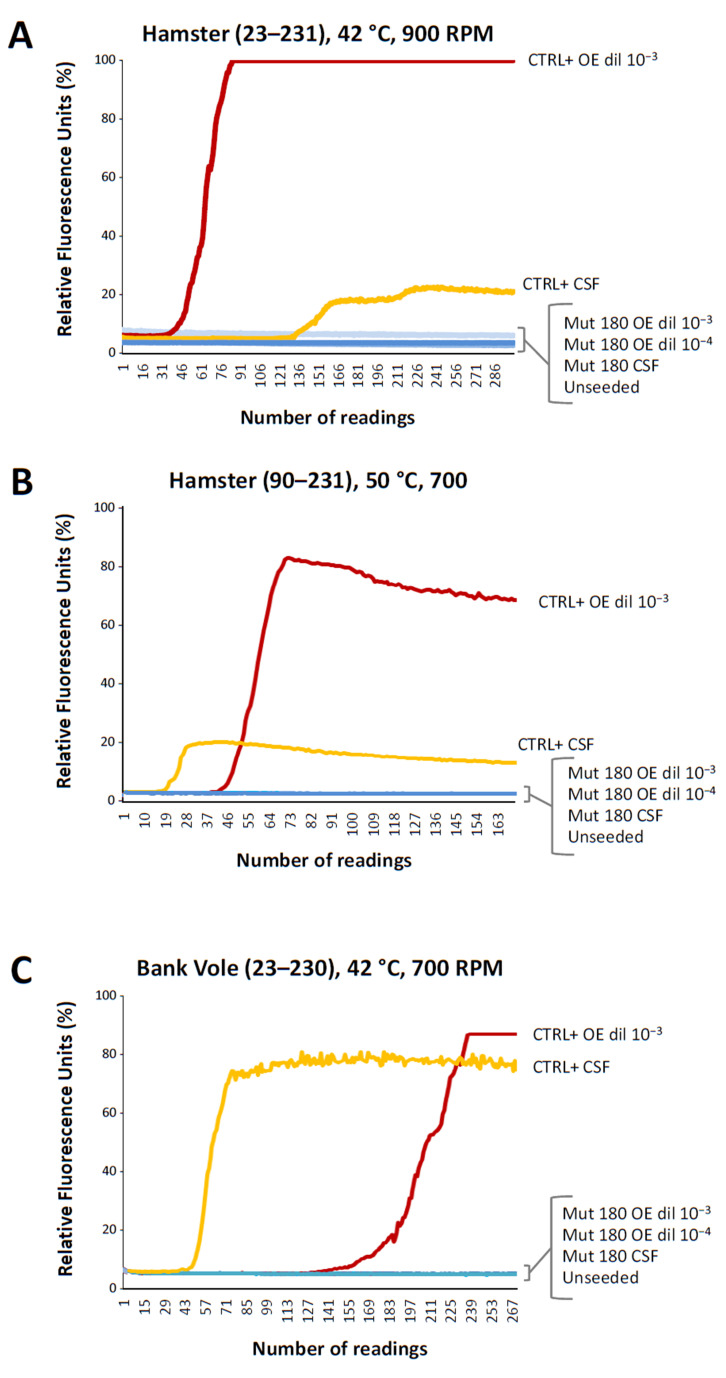
RT-QuIC analyses. RT-QuIC performed on CSF and OE in the presence of the recombinant full-length Syrian hamster prion protein (panel (**A**)), truncated Syrian hamster prion protein (panel (**B**)), and full-length bank vole prion protein (panel (**C**)) under specific conditions of temperature and shaking. Mut 180: samples from V180I patient; CTRL+: samples from definite sCJD; Unseeded: samples containing only the reaction mix. Dilutions of olfactory epithelia used as seeds are reported. Each fluorescence line represents mean fluorescence intensity from 4 replicate wells. Maximum fluorescence level in (**A**–**C**) corresponds to 260,000 relative fluorescence units. OE was collected by nasal brushing.

**Figure 3 ijms-23-10210-f003:**
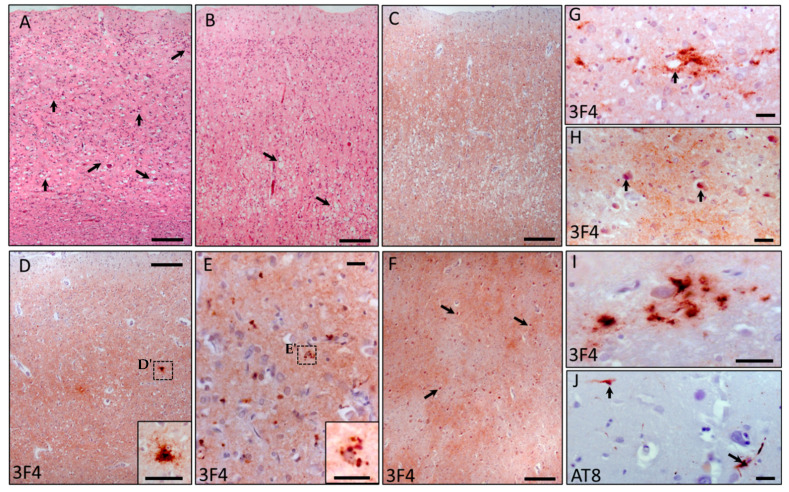
Neuropathological findings. (**A**) Frontal cortex (H&E), prominent reactive gliosis and spongiosis (arrows) (bar 50 µm); (**B**) occipital cortex (H&E), severe spongiform changes characterized by different sizes of non-confluent vacuoles (arrows), particularly in the deeper layers (bar 50 µm); (**C**) occipital cortex (PrP immunostaining), synaptic-like deposition (bar 50 µm); (**D**) entorhinal cortex (PrP immunostaining), intense PrP staining with isolated glial-like deposits (dashed square, (**D’**)) (bar 50 µm; inset bar 25 µm (**D’**)); (**E**) amygdala (PrP immunostaining), intracellular PrP deposits (dashed square, (**E’**)) (bar 25 µm; inset bar 25 µm (**E’**)); (**F**) posterior thalamus, synaptic-like and intracellular PrP staining (arrows) (bar 50 µm); (**G**) entorhinal cortex, PrP perivascular aggregates (arrow) (bar 25 µm); (**H**) posterior thalamus, intracellular PrP staining and occasional aggregates (arrows) (bar 25 µm); (**I**) thalamus, sporadic PrP aggregates; (**J**) hippocampus, occasional tau-positive neurons and neurites (arrows) (bar 25 µm). H&E (panels (**A**,**B**)); PrP immunostaining ((**C**–**I**) with 3F4, 1:1000, (**J**) with AT8, 1:1000).

**Figure 4 ijms-23-10210-f004:**
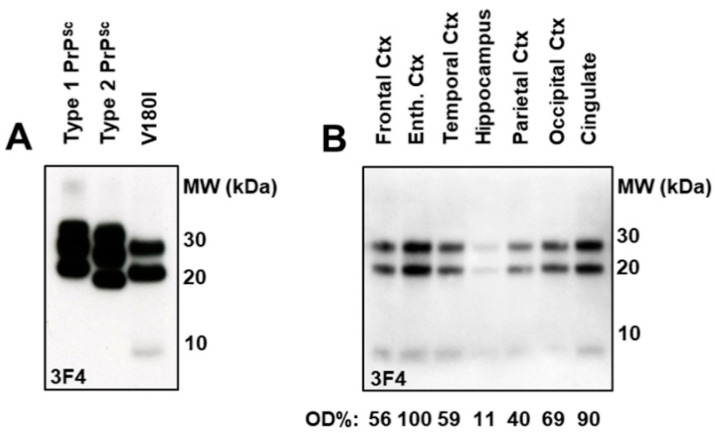
Biochemical characterization and intracerebral distribution of PrP27–30 in the brain of the V180I patient. (**A**) Immunoblot analysis of the brain homogenates from V180I gCJD case and sCJD cases with a 3F4 antibody. In contrast with sCJD brain samples, V180I PrP27–30 lacks the di-glycosylated band and shows the fast-migrating band of 8 kDa (lane 3, left). (**B**) PrP27–30 shows the same pattern in different areas of V180I. High amounts of PrP27–30 are found in the temporal, entorhinal, cingulate and frontal cortexes. Every lane was loaded with 300 μg of proteinase K-digested homogenate. The relative percentage of PrP27–30 is reported as optical density (OD%).

## Data Availability

The data presented in this study are available on request from the corresponding author.

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
