# Peer review of "Biochemical and Neuropathological Findings in a Creutzfeldt–Jakob Disease Patient with the Rare Val180Ile-129Val Haplotype in the Prion Protein Gene"

_ijms, 2022, doi:10.3390/ijms231810210_

Round 1

Reviewer 1 Report

The Manuscript: „ Biochemical and neuropathological findings in a Creutzfeldt- Jakob disease patient with the rare Val180Ile-129Val haplotype in the prion protein gene’’ by Gianluigi Zanusso and colleagues report on a gCJD patient with the rare V180I-129V haplotype, showing an unusual long duration of the disease and a characteristic PrPSc glycotype. Previously, inherited mutations in the Prion protein (PrP), encoded by the PRNP gene, have been known to be associated with various autosomal dominant neurodegenerative disorders, such as Creutzfeldt–Jacob disease (CJD), Gerstmann–Sträussler–Scheinker syndrome (GSS) and Fatal Familial Insomnia (FFI). The present study expands the genetic spectrum of Creutzfeldt- Jakob disease associated with PRNP gene. The study is nicely conducted with elaborate description of methodology and documentation and interpretation of subsequent result. After going through the manuscript, I have few comments for the authors:

1.     What was the age of two family members with V180I mutation and does the genetic results align with mutation segregation in the family?

2.     Is V180I variant listed as a pathogenic mutation in different mutation databases? Is this mutation also identified previously in other neurodegenerative disorders, such as Gerstmann–Sträussler–Scheinker syndrome (GSS) or Fatal Familial Insomnia (FFI)?

3.     The origin of the patient is not clearly mentioned in the manuscript. Is the patient Caucasian?

4.      Please discuss briefly how the origin, penetrance, disease phenotype and transmissibility of 180I-129V haplotype could be further analysed.

5.     I would suggest the authors to include the concluding remarks highlighting the novelty and significance of the study.

Author Response

Response to reviewer 1

ijms-1859294: Biochemical and neuropathological findings in a Creutzfeldt-Jakob disease patient with the rare Val180Ile-129Val haplotype in the prion protein gene

Rev

Comment

Response

1

The Manuscript: „ Biochemical and neuropathological findings in a Creutzfeldt- Jakob disease patient with the rare Val180Ile-129Val haplotype in the prion protein gene’’ by Gianluigi Zanusso and colleagues report on a gCJD patient with the rare V180I-129V haplotype, showing an unusual long duration of the disease and a characteristic PrPSc glycotype. Previously, inherited mutations in the Prion protein (PrP), encoded by the PRNP gene, have been known to be associated with various autosomal dominant neurodegenerative disorders, such as Creutzfeldt–Jacob disease (CJD), Gerstmann–Sträussler–Scheinker syndrome (GSS) and Fatal Familial Insomnia (FFI). The present study expands the genetic spectrum of Creutzfeldt- Jakob disease associated with PRNP gene. The study is nicely conducted with elaborate description of methodology and documentation and interpretation of subsequent result. After going through the manuscript, I have few comments for the authors:

1

1) What was the age of two family members with V180I mutation and does the genetic results align with mutation segregation in the family?

The age of the two family members with V180I mutation was 65 and 45 years. We included this information in the manuscript.

The genetic results align with mutation segregation. Additional details on family members were not described to protect their privacy.

1

2) Is V180I variant listed as a pathogenic mutation in different mutation databases? Is this mutation also identified previously in other neurodegenerative disorders, such as Gerstmann–Sträussler–Scheinker syndrome (GSS) or Fatal Familial Insomnia (FFI)?

According to the National Center for Biotechnology Information. ClinVar; [VCV000013405.9], https://www.ncbi.nlm.nih.gov/clinvar/variation/VCV000013405.9 (accessed Aug. 11, 2022) the SNP rs74315408 (PRNP c.538G>A (p.Val180Ile) is interpreted as pathogenic/likely pathogenic (Missense variation).

There are no reported neurodegenerative disorders, other than CJD, associated to this mutation (NCBI, pubmed). The final diagnosis of all published cases was gCJD. However, the possibility that the mutation is at play with genes associated with other neurodegenerative disorders can not be excluded (Lee SM, Chung M, Hyeon JW, Jeong SW,Ju YR, Kim H, et al. Genomic Characteristics of Genetic Creutzfeldt-Jakob Disease Patients with V180I Mutation and Associations with Other Neurodegenerative Disorders. PLoS ONE 2016;11:e0157540. doi:10.1371/journal.pone.0157540)

1

3) The origin of the patient is not clearly mentioned in the manuscript. Is the patient Caucasian?

The patient was Caucasian. Added in the text.

1

4) Please, discuss briefly how the origin, penetrance, disease phenotype and transmissibility of 180I-129V haplotype could be further analysed.

We added a couple of sentences in the discussion

1

5) I would suggest the authors to include the concluding remarks highlighting the novelty and significance of the study.

We added a “Concluding remarks” section as suggested.

1 + 2

Moderate/minor english changes are required.

Text has been revised.

Reviewer 2 Report

The authors present an interesting study examining the influence of a specific mutation in a prion protein in the context of Creutzfeldt Jakob disease. Utilising clinical data, the association of the mutation with onset of CJD is explored, and while the approach is interesting, it is clear that further work is warranted in determining the true extent of the mutations influence in this context.

In reviewing the manuscript, I had a number of concerns. The following should be addressed when preparing a suitable revision.

1. It would be useful if for the immunohistochemistry data panel that arrows be utilised to point the reader in the direction of the physiological change of relevance/note with respect to this investigation.

2. The methods could use more details. For example, any antibodies that were utilised should also have their product codes listed in addition to the company from which they were purchased.

3. It would be useful if details (type, source, concentration, etc) for the secondary antibodies used through the study were also given.

4. The method for the Western blotting needs to be expanded upon. The level of detail is quite light in its current form, and there are not sufficient details that would allow a reader to replicate this study.

5. The same can be said about the method for RT-QuIC. The authors reference other sources for details on how this procedure is performed, but it is of this reviewers opinion that these details he included relevant to this study, if even briefly.

6. Moreover, the IHC method is also light on details.

7. The formatting of the RT-QuIC data in figure 2 could be improved somewhat. The labels are at times hard to read, and efforts should be made to improve this.

8. For the western blot data, the first is somewhat over exposed. Have the authors any better representative blots?

9. The second western blot exhibits several bands. How can the authors be certain these are the targets of interest and not non-specific bindings?

Author Response

Response to reviewer 2

ijms-1859294: Biochemical and neuropathological findings in a Creutzfeldt-Jakob disease patient with the rare Val180Ile-129Val haplotype in the prion protein gene

Rev

Comment

Response

2

The authors present an interesting study examining the influence of a specific mutation in a prion protein in the context of Creutzfeldt Jakob disease. Utilising clinical data, the association of the mutation with onset of CJD is explored, and while the approach is interesting, it is clear that further work is warranted in determining the true extent of the mutations influence in this context.

In reviewing the manuscript, I had a number of concerns. The following should be addressed when preparing a suitable revision.

2

1) It would be useful if for the immunohistochemistry data panel that arrows be utilised to point the reader in the direction of the physiological change of relevance/note with respect to this investigation.

Figure 3 has been revised where possible.

2

2) The methods could use more details. For example, any antibodies that were utilised should also have their product codes listed in addition to the company from which they were purchased.

Done

2

3) It would be useful if details (type, source, concentration, etc) for the secondary antibodies used through the study were also given.

Done

2

4) The method for the Western blotting needs to be expanded upon. The level of detail is quite light in its current form, and there are not sufficient details that would allow a reader to replicate this study.

Done

2

5) The same can be said about the method for RT-QuIC. The authors reference other sources for details on how this procedure is performed, but it is of this reviewers opinion that these details he included relevant to this study, if even briefly.

Done

2

6) Moreover, the IHC method is also light on details.

Done

2

7) The formatting of the RT-QuIC data in figure 2 could be improved somewhat. The labels are at times hard to read, and efforts should be made to improve this.

Figure 2 has been amended

2

8) For the western blot data, the first is somewhat over exposed. Have the authors any better representative blots?

The blot is over exposed to show the 8kDa fragment that otherwise would not be promptly visible

2

9) The second western blot exhibits several bands. How can the authors be certain these are the targets of interest and not non-specific bindings?

PrP banding pattern of V180I brain samples shown in 4B is identical to those shown in 4A and reproduces the typical banding pattern of V180I disease associated-PrP reported in other studies and different from PrPSc associated with sCJD (figure 4A).

1 + 2

Moderate/minor english changes are required.

Text has been revised.

Round 2

Reviewer 2 Report

The authors have suitably addressed my comments and the manuscript is subsequently much improved